# Laser-Plasma Spectroscopy of Hydroxyl with Applications

**DOI:** 10.3390/molecules25040988

**Published:** 2020-02-22

**Authors:** Christian G. Parigger, Christopher M. Helstern, Benjamin S. Jordan, David M. Surmick, Robert Splinter

**Affiliations:** 1Physics and Astronomy Department, University of Tennessee, University of Tennessee Space Institute, Center for Laser Applications, 411 B.H. Goethert Parkway, Tullahoma, TN 37388-9700, USA; chris.helstern@gmail.com; 2Nuclear Engineering Department, Tickle College of Engineering, University of Tennessee, 1412 Circle Drive, Knoxville, TN 37912, USA; bjordan1@vols.utk.edu; 3Physics and Applied Physics Department, University of Massachusetts Lowell, Lowell, MA 01854, USA; David_Surmick@uml.edu; 4Wellinq Medical, Van der Waals Park 22, 9351 VC Leek, The Netherlands; rsplinter@gmail.com

**Keywords:** plasma diagnostics, molecular spectra, diatomic molecules, plasma spectroscopy, laser spectroscopy, laser-induced breakdown spectroscopy, optical emission spectroscopy, molecular excitation temperature, combustion analysis, astrophysical spectroscopy

## Abstract

This article discusses laser-induced laboratory-air plasma measurements and analysis of hydroxyl (OH) ultraviolet spectra. The computations of the OH spectra utilize line strength data that were developed previously and that are now communicated for the first time. The line strengths have been utilized extensively in interpretation of recorded molecular emission spectra and have been well-tested in laser-induced fluorescence applications for the purpose of temperature inferences from recorded data. Moreover, new experiments with Q-switched laser pulses illustrate occurrence of molecular recombination spectra for time delays of the order of several dozen of microseconds after plasma initiation. The OH signals occur due to the natural humidity in laboratory air. Centrifugal stretching of the Franck-Condon factors and r-centroids are included in the process of determining the line strengths that are communicated as a Supplementary File. Laser spectroscopy applications of detailed OH computations include laser-induced plasma and combustion analyses, to name but two applications. This work also includes literature references that address various diagnosis applications.

## 1. Introduction

Practical applications of molecular spectroscopy include diagnosis of hydroxyl (OH) diatomic molecules in the study of laboratory plasma and chemical processes. This work communicates state-of-the-art OH line-strength data that were well-tested in laser-induced fluorescence and emission spectroscopy diagnoses. Laser spectroscopy [1,2,3] usually delivers information about the temporal- and spatial- distribution of species. Diagnosis usually utilizes methods that are frequently encountered in characterization of plasma [4]. This work also communicates aspects of laser spectroscopy of OH that include active interrogation with excitation from ground- or lower molecular states and levels, viz. laser-induced fluorescence (LIF), and passive analyses of excited molecular states or levels encoded in the emitted light from laser-induced plasma, viz. optical emission spectroscopy (OES). However, laser spectroscopy examples primarily focus on OES of the OH molecule and at standard ambient temperature and pressure (SATP) laboratory conditions.

Molecular recombination spectra are readily observed following optical breakdown in laboratory air. Emission signals from the OH molecule in the UV near 306 nm can be recognized in air breakdown for time delays of the order of 10 μs after cessation of the laser pulse, but recorded data for the Δv=0 OH bands indicate spectral interference with N_2_ second positive bands [5,6]. For time delays of the order of 100 μs, however, ultraviolet OH emissions of the Δv=0 bands dominate in the range of 305 nm to 322 nm. Hydrocarbon combustion typically produces OH that can be measured with laser-induced fluorescence (LIF) and planar laser-induced fluorescence (PLIF) [7,8].

This work aims to (a) publish OH line strength data [9] that can be directly utilized with associated graphing and non-linear fitting programs [10] that include the Nelder-Mead temperature (NMT) program and the Boltzmann equilibrium spectroscopy program (BESP); (b) communicate new experimental investigations for the purpose of estimating both typical signal strength and time-delays for occurrence of hydroxyl spectra in laboratory laser-sparks; (c) summarize possible approaches for determination of OH concentrations with reference to local equilibrium conditions for a variety of plasma species; (d) illustrate typical fluid physics phenomena that can be initially explored by employing shadowgraph techniques; (e) comment on the set extensively LIF- and OES- tested OH line strength data with respect to alternate or perhaps historic approaches for computation of theory spectra, and (f) stimulate further fundamental analysis and comparisons with laboratory data.

Fundamental analysis includes investigations of differences in spectra for OH and OD isotopologues using the program LIFBASE [11]. Applications include laser ablation molecular isotopic spectrometry (LAMIS) [12]. Recent OH data-base developments [13] for predictions with PGOPHER [14] indicate the continued interest in establishing well-defined parameters for computation of OH spectra for analysis of diatomic molecule signatures, viz. hydroxyl, abundant in interstellar medium (ISM) along with other matter and radiation in the region between star systems. In that work, including exo-planet studies, the ExoMol data-base [15,16] is expected to assist in analysis of available superposition spectra.

## 2. Mini-Summary of OH Literature and Applications

The OH diatomic molecule occurs frequently as an intermittent molecule as part of a volley of chemical reactions that occur for example in plasma and that are measured in combustion processes. Examples of fundamental interests include exploration of the effects of chemical reactions on non-isothermal plasma [17] and the effect of N_2_ and O_2_ on OH production [18]. Related studies focus on line-by-line emission spectra from non-equilibrium plasma [19].

In another application, the OH radical has been employed for velocity mapping [20]. The hydroxyl group provides an attractive flow tracer, which includes the use of aldehydes in quantifying the air-fuel ratio in combustion chambers [21,22]. Furthermore, the use of laser-induced fluorescence yields high accuracy in determination of peroxy radicals [23]. The peroxide radial quantification process has particular applications in the determination of removal of pollutants, which are contributing to the oxidation of hydrocarbons. Furthermore, the reaction of peroxy radicals with nitrogen monoxide and -dioxide leads to the formation of ozone, a general environmental concern [23]. Laser-induced fluorescence can for instance be applied to quantify the presence of pollutants in the atmosphere, i.e., troposphere, based on the reactions with OH. The hydroxyl group containing radials are the primary result of photolysis of ozone [23].

Measurement and tracking of OH levels in the tropospheric atmosphere is of critical importance owing to the fact that OH is the most important oxidant according to Wang because of its ability to remove trace gases [24]. Laser Induced Fluorescence has been used as the basis for a ground-based measuring system for tracking the OH levels in the tropospheric atmosphere and this ability will continue to be important and relevant to monitor the pollutants in the atmosphere [24]. Planar laser-induced fluorescence (PLIF) is one of the most widely used techniques for investigating flows of reactants such as OH [25]. Understanding of the temperatures of the reactants in flows has been a subject of recent investigation through the fitting of simulated OH excitations spectra [26,27]. These applications require continued deepened understanding of the behavior of OH plasma as offered in the research discussed in this article.

Additional literature on OH spectroscopy is rather exhaustive. This work however communicates a selected subset of papers that are of interest in hydroxyl diagnosis. In other words, various References are included to convey a mini-review of relevant literature that focuses on the OH diatomic molecule. Spectroscopic data for OH analysis are typically derived from Fourier transform spectroscopy [28,29] or by detailed studies of rotational spectra [30]. Fundamental data of OH molecular transitions are communicated in selected references [31,32,33,34,35,36,37,38,39,40,41,42,43,44,45,46,47,48,49]. Interrogations of ground-state or ground-level populations employ laser-induced fluorescence [50,51,52,53,54,55,56,57,58,59,60,61,62,63,64,65,66,67,68] with practical applications to combustion diagnosis [69,70,71,72,73,74]. However, diagnosis applications are based on extensive studies of the ultraviolet OH system [75,76,77,78,79,80,81,82,83,84,85,86,87,88,89,90,91,92,93].

## 3. Experimental Details

Standard experimental components are used for laser-induced breakdown spectroscopy, summarized previously, but are included for completeness, e.g., see reference [94] or for general laser-induced breakdown spectroscopy references [95,96]. The arrangement for the experiments reported in this work consists of a set of components typical for time-resolved, laser-induced optical emission spectroscopy, or nanosecond laser-induced breakdown spectroscopy (LIBS) [95]. Primary instrumentation includes a Q-switched neodymium-doped yttrium aluminum garnet, Nd:Y_3_Al_5_O_12_ (Nd:YAG) device (Quantel model Q-smart 850, USA) operated at the fundamental wavelength of 1064-nm to produce full-width-at-half-maximum 6-ns laser radiation with an energy of up to 850 mJ per pulse, a laboratory type Czerny-Turner spectrometer (Jobin Yvon model HR 640, Fr) with a 0.64-m focal length and equipped with a 1200 grooves/mm grating, an intensified charge coupled device (Andor Technology model iStar DH334T-25U-03, USA) for recording of temporally and spatially resolved spectral data, electronic components for synchronization, and various optical elements for beam shaping, steering and focusing.

In previous experiments, captured shadowgraphs of the breakdown plasma [97] served the purpose of visualizing the plasma expansion when using 850 mJ, 6-ns radiation. The captured images are consistent with results of fluid-dynamic expansion phenomena [98] presented in the literature. In laser-ablation research, shockwave expansion studies [99] from a solid sample into air or a into a gaseous environment such as argon are important for laser-induced breakdown spectroscopy (LIBS) that is applied for determination of elemental composition of the sample.

However, it is important to obtain shadowgraphs for plasma excitation energies that were employed for time-resolved spectroscopy. Shadowgraphs reported recently [94] are captured using two separate laser devices (Continuum Surelite model SL I-10, USA) that can be externally operated to deliver laser pulses with a well-defined time delay showing less than ±1 ns trigger-jitter between the pulses. The experiments reported in reference [94] are extended to time delays of the order of 100 μs for the purpose of visualizing the breakdown site for time delays that correspond to reasonable OH signals.

For visualization studies, both lasers are frequency-doubled to operate at the 2nd harmonic, 532-nm wavelength, and both beams are spatially overlapped. Both pulses can be delivered with a minimum time delay of 300 ns. Shadowgraphs are recorded by external synchronization of the Surelite and Quantel laser devices and by externally triggering the camera (Silicon Video 9C10 color camera, USA) that records the images that are projected onto a screen.

## 4. Results and Discussion

### 4.1. Shadowgraphs

Investigations of expanding laser-induced shockwaves and fluid-physics phenomena utilize effectively high shutter-speed shadowgraph photography. Figure 1 illustrates the essential experimental module that is composed of a focusing lens for generation of the optical breakdown plasma, and an imaging lens that projects the interaction volume onto a screen. Two adjustable iris are employed for alignment. The iris on the laser side allows on to adjust the ratio of focal length, F, and of the beam diameter, D. For the shadowgraph studies reported in this work, a ratio of F/D = f^#^ = 20 is used.

Figure 2a,b illustrate typical shadowgraphs recorded for time-delays of 54.25 μs and 104.25 μs, respectively. The laser is incident from the right as indicated by the arrow in the single shot captured shadowgraph. While an outgoing shockwave occurs for time delays of the order of 1 μs, as recently communicated in reference [94], Figure 2 depicts well developed vortices and fluid flow towards the incoming laser beam. In view of laser spectroscopy, a time-resolved data taken along a narrow slice along the horizontal direction would be affected by fluid dynamics.

### 4.2. Measurement of OH Spectra

Time-resolved spectroscopy of laser-induced plasma in laboratory air at a relative humidity level of 25% reveal characteristic ultraviolet OH data due to molecular recombination. The natural moisture present in the air causes occurrence of OH spectra. In thermodynamic equilibrium, the chemical composition of plasma is comprised of a volley of species. Figure 3 illustrates computed OH density versus temperature using the chemical equilibrium with applications (CEA) program [100,101].

For relative humidity in the range of 5% to 50%, the maximum OH signals are in the range of 3000 K to 3200 K. At a humidity level of 25% and at a temperature of 293.15 K (21 °C), air contains water at an amount of 4 g/kg_dryair_ [102], or 0.0114 mol/mol_dryair_ [102]. The dry air composition [103] and the H_2_O content provide the input parameters for the CEA program [101] are listed in Table 1. Minor species such as carbon dioxide, helium, methane, krypton, hydrogen, xenon show mole-fractions below $2\times 10^{-5}$ and are not included in the computations, except hydrogen. The hydrogen mole fraction in dry air is of the order of $5\times 10^{-6}$. This residual amount of hydrogen cannot cause the level of OH signals reported in this work. In fact, CEA results obtained with 50% relative humidity show 1000× larger maximum OH density (near 3200 K) than that for dry air (near 2200 K) containing a mole fraction of $5\times 10^{-6}$ hydrogen.

Optical breakdown was generated at a rate of 10 Hz, with the laser beam focused with f/5 optics from the top, or parallel to the slit, analogous to recently reported CN laser spectroscopy [94]. Figure 4 illustrates the experimental arrangement for imaging of the laser-plasma onto the vertical slit of the spectrometer. The laser beam, parallel to the slit, arrives from the top (red arrow) and is focused with f/5-optics by an anti-reflection coated, 25.4-mm (1 in.) lens. The air-breakdown plasma, indicated by the asterisk is imaged 1:1 with a 50.8-mm lens.

The spot radius, $\omega_{0}$, for perfect Gaussian-beam focusing [104,105] amounts to
(1)$$\omega_{0}=\frac{2}{\pi }\;f^{\#}\;\lambda_{\mathrm{laser}}\;.$$

With the wavelength, $\lambda_{\mathrm{laser}}=1064$ nm, one finds $\omega_{0}=3.4\;\mu$m. However, spherical aberrations increase the spot-size. Computations of focal volume distributions are presented, for example, in references [6,9,12,106]. Moreover, optical breakdown can occur prior to reaching the focal spot predicted with Gaussian-beam focusing as irradiance-levels reach breakdown threshold [8]. In the experiments, the irradiance is of the order of 10× more than that needed for optical breakdown in dry air.

The detector pixels are binned in 4-pixel tracks along the slit direction, resulting in obtaining 256 spectra for each time delay. Measurements comprise accumulation of 100 consecutive laser-plasma events for 11 separate time delays at 10 μs steps. The selected series explores the plasma decay with specific attention to recognition of OH molecular data free from spectroscopic interference.

Figure 5 and Figure 6 illustrate spatio-temporal spectra that were recorded along the line-of-sight. The slit-height corresponds to the x-direction in Figure 2 and the line-of-sight corresponds to direction of the back light used for capture of shadowgraphs. The images are individually scaled and pseudo-colored using a rainbow distribution.

The raw spectra in Figure 5 and the corresponding averages in Figure 6 are consistent with previously recorded spectra using a UV-enhanced, intensified linear-diode array [5,107]. Figure 6 shows the averages of the recorded counts. Early delay-time data were analyzed using classic laser spectroscopy, namely, addition of superimposed N_2_ second positive and OH spectra. Moreover, a program designed for study of plasma torches was applied using chemical equilibrium. It is clear that a variety of species, including electrons, contribute to the measured spectra [5,6]. Investigations of information content and error propagation analysis reveals reasonable diagnosis capabilities of the line-of-sight analysis.

In view of the recorded shadowgraphs (see Figure 2), there appears to be well defined fluid physics phenomena including vortex formation and implosion of the shockwave-dominated expansions. The vortex is well developed and the implosion towards the laser device is recognizable in Figure 2 for time delays of approx. 50 and 100 μs. The direction of the implosion is determined by slightly asymmetric energy absorption, in other words, by the direction of the laser-beam. The energy absorption causes slight asymmetries of shock waves for pulse energies of 160 to 200 mJ/pulse [5], and the expansion speeds are above hypersonic for time delays of approx. 0.5 μs.

Table 2 conveys the shockwave radii for laser-induced plasma initiated with energies 200 mJ/pulse in standard ambient temperature and pressure (SATP) air. The table entries confirm that for 200-mJ/pulse the shockwave is outside the vertical dimension of the available slit of approx. 14-mm height for 1:1 imaging. The shockwave radii were calculated using Taylor-Sedov model, analogous to recently published investigation of CN spectra [94],
(2)$$R\left(\tau \right)={\left(\frac{E}{\rho }\;\tau^{2}\right)}^{1/5}.$$

The shockwave radius, R(τ), is a function of absorbed pulse energy, *E*, density of the gas, ρ, and time delay, τ.

### 4.3. Computation of OH Spectra

The computation of OH diatomic molecular spectra is based on collection of line-strength data [9] and application of algorithms for comparison of measured and computed spectra [10]. Figure 7 illustrates computed OH data for spectral resolutions of 0.25 nm and 0.0025 nm that correspond to typical resolutions in laser-induced spectroscopy with intensified array detectors and nominal pico-meter resolution stick-spectra. The line-strength data are included as a Supplement, and Appendix B reports typical fitting results of well-calibrated data [5] in terms of wavelength and instrument sensitivity.

### 4.4. Comparisons of Recorded and Computed OH Spectra

The accuracy of the line strength data can be evaluated by comparing predictions with optical emission spectroscopy (OES) and laser induced fluorescence (LIF) records. Line strength wavelengths are accurate to better than 0.05 cm^−1^ or better than 0.5 pm at a wavelength of 300 nm. In passive emission spectroscopy, small variations in the precision of the computed spectral data can add up to differences in predicted and measured data although laser-plasma spectral resolutions are frequently of the order of 0.1 nm, in other words, the line strength wavelengths are known about a factor of 200 better than available in OES. In addition, the predicted relative magnitudes of individual lines within a band system need to agree as well.

The hydroxyl A ^2^$\Sigma^{+}↔$ X ^2^$\Pi_{i}$ ultraviolet system line strengths (see Appendix B) contain 0-0, 0-1, 1-0, 1-1, 1-2, 2-0, 2-1, 2-2, and 2-3 bands. Centrifugal stretching of the Franck-Condon factors and r-centroids are included in the process of determining the line strengths. Figure 8 and Figure 9 illustrate laser induced fluorescence and optical emission spectroscopy comparisons. The specific details of the LIF and OES experiments conducted for measurement results included in Figure 8 and Figure 9 are not described in detail, important are demonstrations of the line-strength data accuracy and OH diatomic molecular band appearances. Figure 9 also shows contributions from the different bands of the Δv=0 OH transitions.

### 4.5. Temperature and Density of OH

The temperature is inferred from the OH spectra by fitting measured with computed molecular spectra. The procedure, program and a example result are included in the Appendix B. In this work, an average temperature is obtained from the line-of-sight data assuming an equilibrium plasma excitation temperature. This particular assumption has been investigated by employing predictions from chemical equilibrium codes as input to a general non-equilibrium air radiation program [5,6]. Various species densities are predicted within the equilibrium assumption, and include OH, Ar, N, O, e^−^, N_2_, O_2_, CO, NO, NO^+^ (see Table 1 in Reference [5]). For laboratory air, the OH concentration maximum occurs at temperatures of approx. 3000 K at a time delay of approx. 100 μs and are of the order of 10^16^ cm^−3^. A detailed study [5] used linear diode arrays and employed up to 100 Hz repetition rate from a Nd:YAG laser device (Coherent Infinity 40-100, USA). Determination of OH temperature and molecular species concentrations in laser-induced laboratory air breakdown includes Monte Carlo simulation as well for estimation of error bars [5]. Contributions from species other than OH for time delays of the order of 100 μs can be calculated using a general non-equilibrium program in the equilibrium mode, with inputs obtained from chemical equilibrium codes - one finds approx. 5% wavelength-dependent peak-background contributions from species other than OH [108,109].

In view of the shadowgraphs reported in this work (see Figure 2), averages of well-developed fluid physics phenomena were recorded with linear diode arrays, viz. summing along the slit-height. It would appear reasonable to determine an association of spatially resolved spectra with the recoded expansion. However, simultaneous recording of shadowgraphs and spectra would be necessary for establishment of expansion characteristics following laser-induced breakdown in air. Such an association of shadowgraphs for time-delays of the order of 1 μs has been reported recently using CN emission spectroscopy [94]. Radon inverse transforms or computed tomography would appear necessary, but possibly Abel inverse transforms may be sufficient as one realizes the symmetry of the vortex indicated in Figure 2.

## 5. Conclusions

This work illustrated analysis of recorded time-resolved OH spectra using accurate line strength files. Presence of the hydroxyl radical is usually associated with combustion, however in laboratory air that contains residual moisture, OH emissions occur as a result of recombination radiation. Laser-plasma and laser-induced fluorescence measurements indicate the accuracy of the OH line strength data that are made publicly available for the first time in this publication. Hydroxyl ultraviolet system line strengths complement those for aluminum monoxide, cyanide, diatomic carbon, and titanium monoxide already available for selected electronic transitions. Future work should address association of OH signals with spatial variations due to vortex-formation and implosions that can be visualized by employing effectively high shutter-speed shadowgraph photography that is only limited by the pulse-width of the back light. The presented work is applicable to electrical-spark analytical chemistry. Moreover, comparisons of PGOPHER and of the nonlinear NMT predictions should be content of future research, including comparisons with LIFBASE predictions. These comparisons are expected to lead to further insights in the analysis of laboratory laser-plasma and on an astronomical scale help advancements in exo-planet and interstellar-medium research.

## Figures and Tables

**Figure 1 molecules-25-00988-f001:**
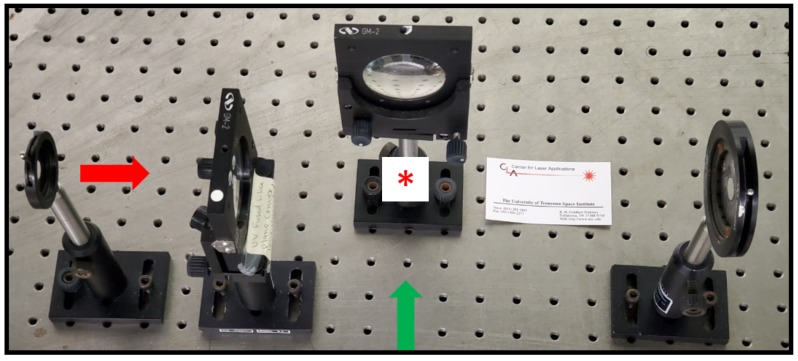
Module for recording shadowgraphs of optical breakdown in air. The collimated laser beam (red arrow) is focused with f/20 optics. The asterisk symbolically indicates optical breakdown. The interaction area is illuminated by a time-delayed laser beam (green arrow), and images are projected onto a screen (not shown) and recorded with a digital camera.

**Figure 2 molecules-25-00988-f002:**
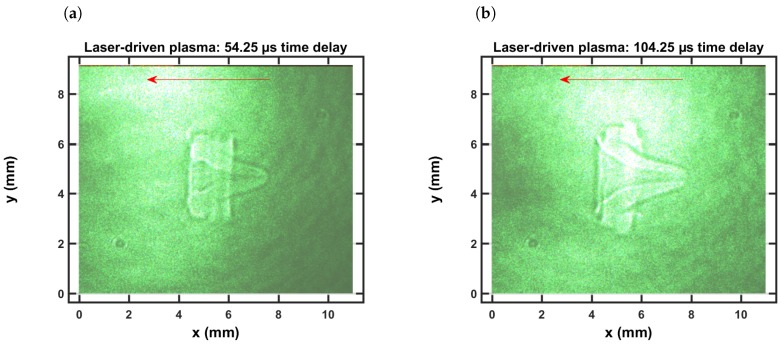
Single-shot shadowgraph of the expanding laser-induced plasma initiated with a 170-mJ,
6-ns, 1064-nm focused beam, and imaged using a 5-ns, 532-nm back-light that is time-delayed by (**a**) 54.25 μs and (**b**) 104.25 μs.

**Figure 3 molecules-25-00988-f003:**
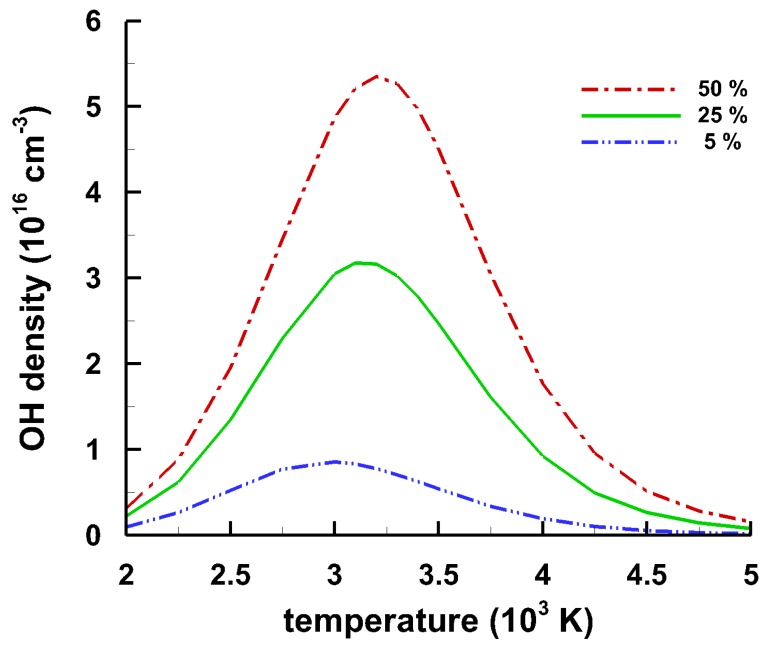
Calculated equilibrium hydroxyl (OH) density versus temperature. For air with 25% relative humidity, the OH density reaches a maximum of ≃3 $\times 10^{16}$ cm^−3^ at T ≃ 3.1 kK.

**Figure 4 molecules-25-00988-f004:**
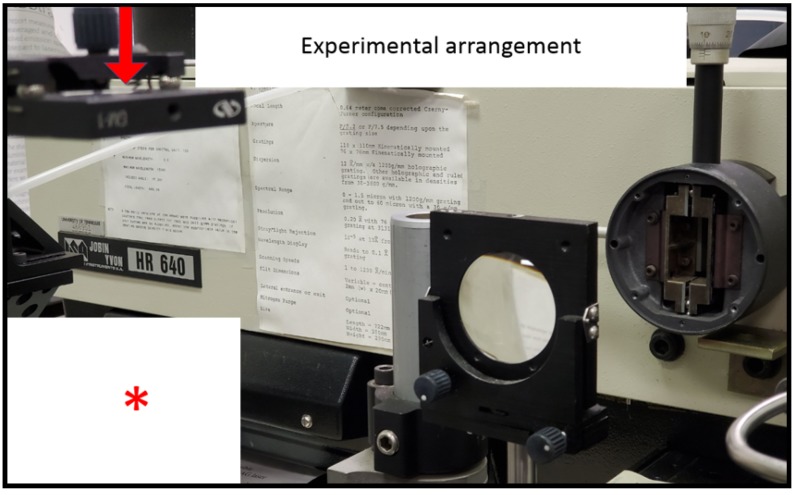
Experimental arrangement for time-resolved spectroscopy. The laser beam (red arrow) is focused to generate optical breakdown, indicated by the asterisk in the lower left of the photograph. The plasma is 1:1 images onto the slit of the spectrometer. Time-resolved data are recorded with an intensified charge-coupled device (not shown) positioned at the exit plane of the spectrometer.

**Figure 5 molecules-25-00988-f005:**
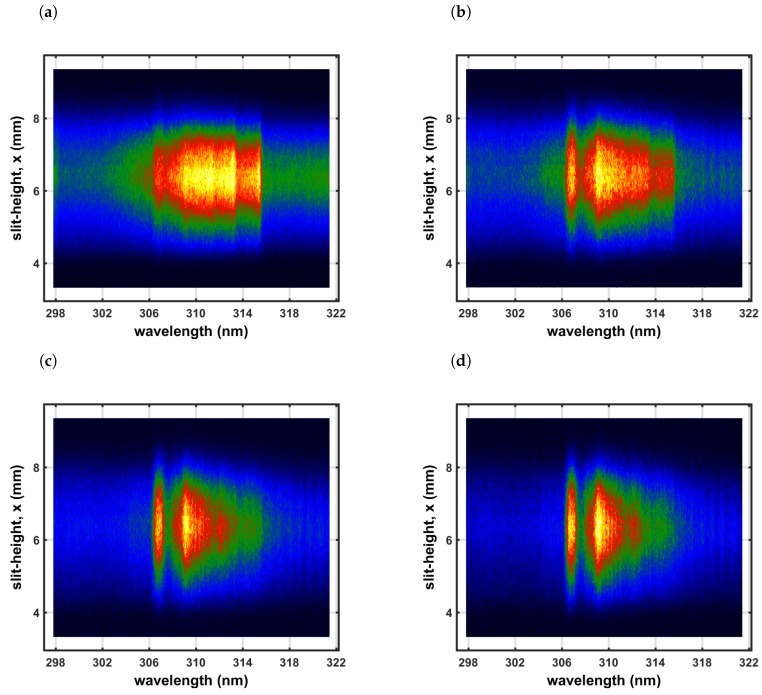
Recorded data of slit-height vs. wavelength. Gate width: 10 μs, time delay (**a**) 20 μs—primarily N_2_ second positive spectra, (**b**) 30 μs, (**c**) 40 μs, and (**d**) 50 μs—primarily OH spectra. Each displayed image is scaled from minimum to maximum and pseudo-colored using a rainbow distribution.

**Figure 6 molecules-25-00988-f006:**
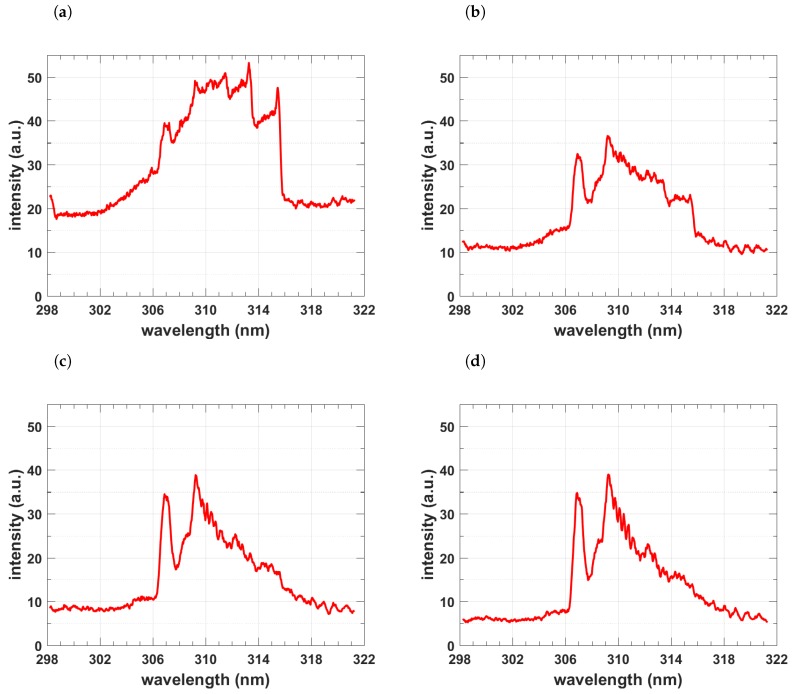
Average of spectra between 4.2 and 8.2 mm of Figure 5. Gate width: 10 μs, time delay (**a**) 20 μs—primarily N2 second positive spectra (**b**) 30 μs, (**c**) 40μs, and (**d**) 50 μss—primarily OH spectra. Contributions from the N_2_ second positive system diminish with increase of time delays.

**Figure 7 molecules-25-00988-f007:**
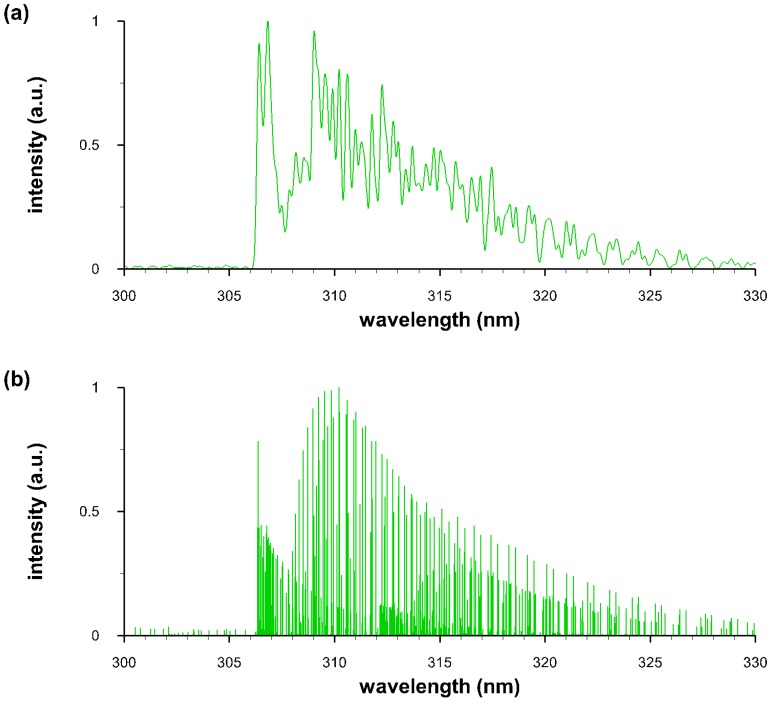
Computed spectra: T = 6000 K; spectral resolution (**a**) 0.25 nm and (**b**) 0.0025 nm.

**Figure 8 molecules-25-00988-f008:**
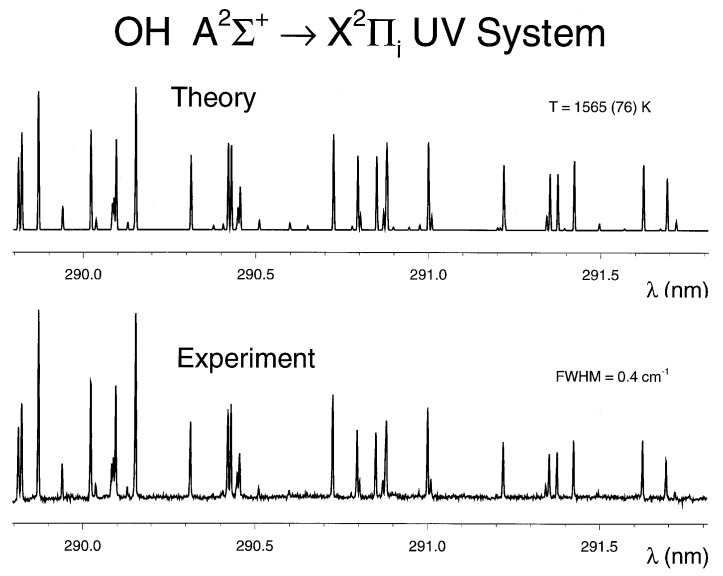
Comparisons of measured and recorded laser induced fluorescence spectra [108].

**Figure 9 molecules-25-00988-f009:**
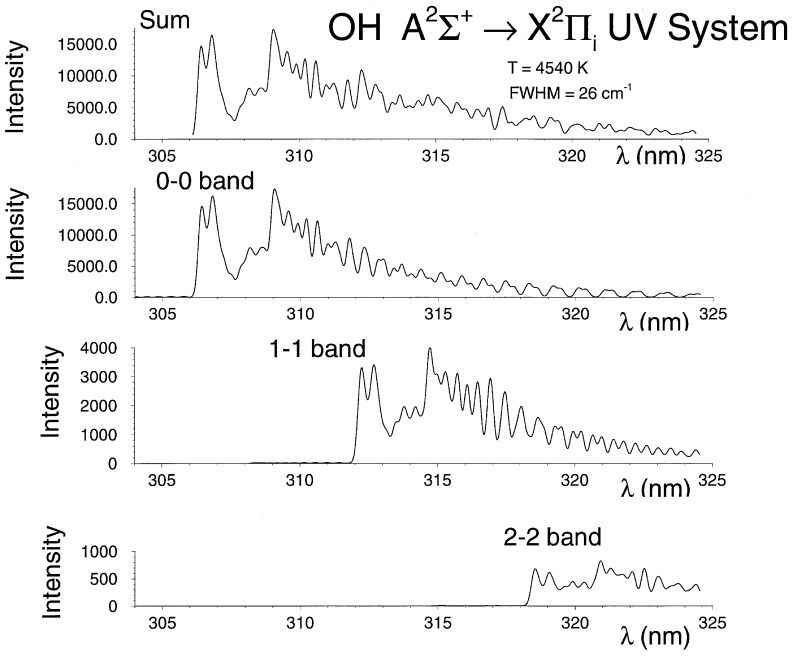
Comparisons of measured and recorded laser-plasma emission spectroscopy [108].

**Table 1 molecules-25-00988-t001:** Input values for computation of air species with the chemical equilibrium with applications (CEA)-code for 25% relative humidity. The sum of mole fractions are normalized to 1 in the code.

Species	Mole Fraction
nitrogen	0.78
oxygen	0.21
argon	0.00934
hydrogen	0.000005
water	0.01144

**Table 2 molecules-25-00988-t002:** Computed shockwave radii for standard ambient temperature and pressure (SATP) air, 200 mJ energy/pulse.

Time Delay (μs)	*R* (mm) for Air [ρ=1.2 kg/m^3^]
20	9.21
25	10.1
50	13.3
75	15.7
100	17.5

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
