# Peer review of "Laser-Plasma Spectroscopy of Hydroxyl with Applications"

_molecules, 2020, doi:10.3390/molecules25040988_

Round 1
Reviewer 1 Report
The paper is a continuation of earlier work with focus on the OH molecule. Results seem to be reliable and should be published.
Some improvements should be made:
1 It should be emphasized in the abstract, in the introduction and also in section 3 that the occurance of OH after plasma formation in air is due to the natural humidity (atomization of h2O molecules)
2 What I miss is a discussion on the intensity of the OH spectra versus humidity
3 I see no reason for using a black background in Fig. 5
Author Response
The manuscript is edited according to the reviewer’s suggestions, and the edits are highlighted with green text.
- Added in abstract and in Section 4.2 – Measurement of OH spectra. that occurrence of OH is due to air humidity. Also included new computations and references.
- Added a discussion of OH density, including a new figure 3, and a new table1. Included new references [103-105].
- Changed the background of the LIF and OES images (now Figs. 8 and 9)
Reviewer 2 Report
The paper presents a time and space resolved spectroscopic sydy of the OH evolution in a laser generated plasma. The results are successfully compared with computer generated spectra.
The paper is well written and in my opinion is of interests to the readers of Molecules.
I have only some minor points for the Authors:
1) It could be a good idea to put a sketch of the experimental apparatus employed for the measurements.
2) Is not clear if the laser was focused in air in front of the HR640 slit. Please make it clear.
3) Please report the focal length of the adopted lens and the resulting focal spot radius.
3) Fig.5 and Fig6 are reproduced on a black background, no problem for the online version, but they are unreadble when printed in B&W.
Publish after minor revisions.
Author Response
The manuscript is edited according to the reviewer’s suggestions, and the edits are highlighted with purple text.
- Included a photograph to show the essential experimental module for shadowgraph-recording, see Fig. 1
- Made clear how the laser beam was focused and how the plasma was imaged by including a photograph, see Fig. 4.
- Included that f/5-optics was used. Included a value for the spot-size. Also added Refs. [106,107].
- Changed the background of the LIF and OES images (now Figs. 8 and 9)
Reviewer 3 Report
The paper by Christian et al. investigate the OH with laser-plasma spectroscopy. The work is very fundamental and important. However, there remains many issues to be more clear. So I can’t recommend the acceptance for the current form till my concerns below have well been responded:
-The paper title is ‘Laser-plasma spectroscopy of hydroxyl with applications’. But I can’t see any possible applications involved. Please revise or add content for the application.
-In the abstract, all the contents are general description. Please summarize some key points contributed from your study in here.
-The motivation is unclear. Why you perform this work? What is the state for detection of OH? Why use your technique? Especially, there have already been some reports as your referred [13-17]. Especially, what is the key points from your summary? You cited a lot of literatures. All of them are related with your work?
- A lot of data are missed enough discussion and comparison like Fig. 1a and 1b, Fig. 5, and Figure 6.
-Fig. 2 does not indicates the meaning of the color, red, green or blue. If not, the meaning will not be clear.
-Some abbreviations are not well defined.
-The English is causal at some places. Please check it thoroughly and improve it further.
Author Response
The manuscript is edited according to the reviewer’s suggestions, and the edits are highlighted with blue text.
- Edited the abstract to include “key”-points, specifically that communication of the well-tested OH line-strength file is communicated publicly. Also added a sentence in the introduction.
- Added two paragraphs in the mini-summary of OH literature and applications. Added applications and references.
- We hope to have made clear that this work discusses various diagnosis techniques, but the comparisons of recorded and computed data is based on the communicated line-strength data. However, we would like to keep the literature review as it also implies the vast application area of OH laser spectroscopy.
- Included a sentence regarding Fig. 5 to indicate image scaling and choice of color-scale. Also added that figure 6 shows the average of the recorded counts for each laser-plasma event in the indicated region.
- Added above Figs. 8 and 9, that these figures are included to highlight the accuracy of the line-strength data for comparison of measured and computed LIF and OES data.
- Also added new references [21-24, 26] and experimental details for the reported, new experiments (also in response to Reviewers 1 and 2 comments)
Round 2
Reviewer 3 Report
All my concerns have well been responded and I recommended the acceptance for the publications.